# Self-Mixing Model of Terahertz Rectification in a Metal Oxide Semiconductor Capacitance

**Fabrizio Palma** 

Rome Università La Sapienza, Dipartimento di Ingegneria dell'Informazione, Elettronica e Telecomunicazioni, 00184 Roma, Italy; fabrizio.palma@uniroma1.it

**Abstract:** Metal oxide semiconductor (MOS) capacitance within field effect transistors are of great interest in terahertz (THz) imaging, as they permit high-sensitivity, high-resolution detection of chemical species and images using integrated circuit technology. High-frequency detection based on MOS technology has long been justified using a mechanism described by the plasma wave detection theory. The present study introduces a new interpretation of this effect based on the self-mixing process that occurs in the field effect depletion region, rather than that within the channel of the transistor. The proposed model formulates the THz modulation mechanisms of the charge in the potential barrier below the oxide based on the hydrodynamic semiconductor equations solved for the small-signal approximation. This approach explains the occurrence of the self-mixing process, the detection capability of the structure and, in particular, its frequency dependence. The dependence of the rectified voltage on the bias gate voltage, substrate doping, and frequency is derived, offering a new explanation for several previous experimental results. Harmonic balance simulations are presented and compared with the model results, fully validating the model's implementation. Thus, the proposed model substantially improves the current understanding of THz rectification in semiconductors and provides new tools for the design of detectors.

**Keywords:** semiconductor device modeling; detectors; terahertz radiation

---

## 1. Introduction

The low-cost detection of high-frequency electromagnetic radiation, particularly terahertz (THz) radiation, using integrated commercial electronics represents a challenging task that is pushing significant experimental and theoretical activities. Novel technologies promise to achieve resolved images that may eventually be colored by the interaction with material chemical bonding.

The radiation spectrum of interest covers practically the entire gap between the microwave and infrared regions. As THz radiation is nonionizing and the associated power is low, it is considered safe. THz wavelengths are able to deeply scan the material under investigation. The combination of these safety and penetration characteristics is important in different applications, such as medical imaging, security/surveillance imaging, and spectroscopic applications.

Recently, significant efforts have been focused on achieving THz sensors using the standard, low-cost, complementary metal oxide semiconductor (MOS) technology, which is characterized, in particular, by strong reproducibility over a large area. This characteristic allows for large arrays or panels of detectors for a large-area detection approach.

Until now, the explanation for the high-frequency detection achieved by MOS field-effect transistors (FETs) [1–3] has been based on plasma wave detection theory [4]. When a high-frequency signal is applied between the gate and source electrodes of a MOS-FET, the model describes how THz radiation generates waves of carriers in the 2D electron gas of the inversion layer. Nonlinearities in the semiconductor equations convert these oscillations into a DC voltage. To increase the detector

responsivity, this approach indicates that a strong downscaling of the gate length is necessary [5]. Nevertheless, for ultra-scaled technology nodes, the parasitic capacitance of the device significantly influences the detector responsivity [6].

Recently, a new approach to the self-mixing process has been developed [7], and this has been applied to the study of the double-barrier structure suitable for high-frequency radiation detection [8]. A preliminary indication of a possible application to the FET structure has been given in [9]. Following a definition that will be justified later in this paper, this approach can be specifically indicated as self-mixing in a drift–diffusion equation, significantly improving the analytical treatment of the self-mixing effect. The model shows how self-mixing occurs extensively in semiconductor regions where a potential barrier is present. In this study, the model is extended to the case of the single barrier present in an MOS capacitance under depletion conditions, below the channel and toward the substrate.

This study considers a unidimensional MOS structure with constant doping of the substrate. The presence of source and drain contacts is not considered; a description of the 2D mechanisms that lead to the appearance of a DC potential at the source and drain electrodes is left for future work.

The proposed model is based on the hydrodynamic semiconductor equations solved for the small-signal approximation. The model depicts the THz modulation mechanisms of the charge in the potential barrier below the oxide and explains the self-mixing process, frequency dependence, and detection capability of the structure. The dependence of the rectified voltage on the bias gate voltage, substrate doping, and frequency is also clarified.

The model is verified through Technology Computer-Aided Design (TCAD) simulations. TCAD solves the semiconductor equations in the structure using, in particular, the hydrodynamic equations. Harmonic balance analysis provides the distributions of all the semiconductor quantities inside the structure.

## 2. Analytical Model

In Figure 1 we present a sketch of the MOS capacitance reporting only the depleted region and the oxide. A substrate of thickness $L_{sub}$ is assumed but is not drawn here. Body contact should be at the bottom of the substrate. Nevertheless, since we will assume that equilibrium of carrier occurs at the edge of the depletion layer and that the resistance of the substrate is negligible, we drew, conventionally, the contact at the edge of the depletion layer. The "abrupt edge" depletion approximation is assumed in the depleted region. In order to simplify the analytical formulation, the depletion region, with dimension $w$, starts at x = 0. At the same edge, it is assumed that Ohmic contact (body) occurs. The oxide region has a thickness of $d_{ox}$. The oxide is assumed to have an equivalent thickness, i.e., $d_{eq} = d_{ox} \frac{\epsilon_s}{\epsilon_{ox}}$. The metal gate contact is assumed to occur at the external side of the oxide region. A polarization voltage $V_G$ is assumed between the gate and the body.

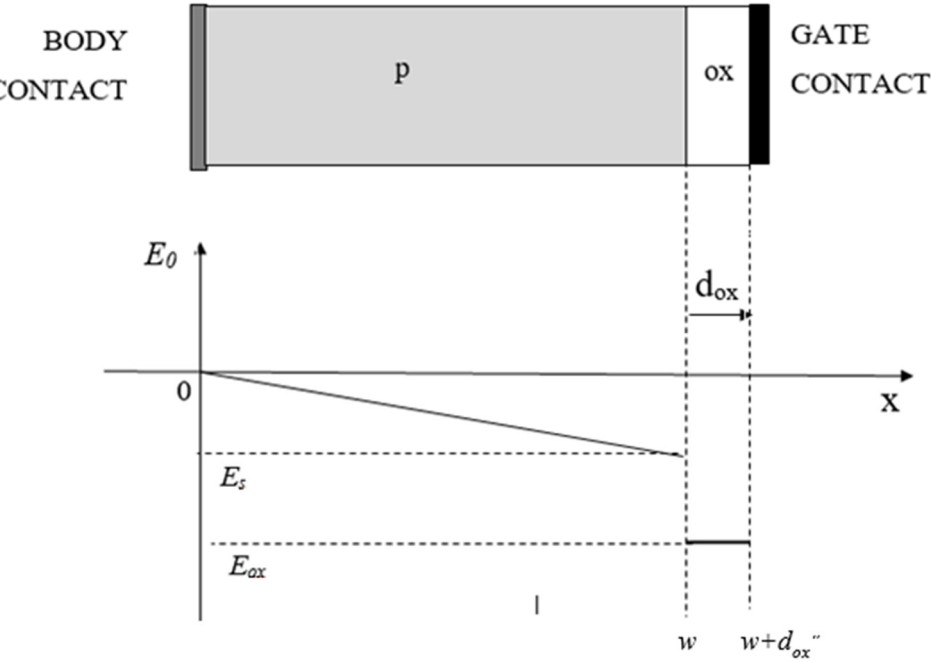

**Figure 1.** The Semiconductor structure and the sketch of the electric field distribution.

Until the inversion condition is reached, the following balance equation can be imposed on the potential:

$$V_G = \frac{1}{2}\frac{q}{\epsilon_s}N_A\,w^2 + \frac{q}{\epsilon_{ox}}N_A w d_{ox} \tag{1}$$

where $N_A$ is the density of the acceptor in the p bulk region, $d_{ox}$ is the thickness of the oxide layer, $\varepsilon_s$ and $\varepsilon_{ox}$ are the dielectric constants of the semiconductor and the oxide, respectively, and $q$ is the absolute electron charge. Equation (1) can easily be solved to obtain $w$. Assuming zero potential at x = 0, within the semiconductor, the potential varies as:

$$\phi(x) = \frac{1}{2}\frac{q}{\epsilon_s}N_A\,x^2 \tag{2}$$

Assuming the thermal equilibrium condition for the carriers, the equilibrium hole density is given by:

$$p_0(x) = p_{p0}e^{-\frac{\phi(x)}{V_T}} \tag{3}$$

where $p_{p0}$ is the equilibrium hole density in the bulk region, which is equal to the doping value $N_A$, $V_T = \frac{k_B T}{q}$, $k_B$ is the Boltzman constant, and $T$ is the absolute temperature.

### 2.1. Radiofrequency Equations

In this model, only a moderate depletion of the semiconductor, albeit without reaching the inversion, is considered, and thus only the majority carriers are considered. The following set of equations can be written as a formulation of the hydrodynamic equations:

$$\frac{\partial\phi}{\partial x} = -E \tag{4}$$

$$\frac{\partial E}{\partial x} = \frac{q}{\varepsilon_S}(p - n - N_A) \tag{5}$$

$$\frac{dp}{dt} = -\frac{\partial p v_p}{\partial x} + D_p\frac{\partial^2 p}{\partial^2 x} \tag{6}$$

$$\frac{\partial v_p}{\partial t} = -v_p \frac{\partial v_p}{\partial x} + \frac{q}{m_p}E - \frac{v_p}{\tau_p} \qquad (7)$$

corresponding respectively to the equations of Poisson, Gauss, continuity of holes, and Euler. Here, $p$ is the total hole density (steady state plus variations), and $v_p$ is the total hole velocity.

The equations adopted are basically the same as those used in [1]. Note that in Equation (6), the contributions from both the transport and diffusion terms are considered. This follows the formulation of the hydrodynamic equation derived in [10] from the Boltzmann equation. This term is also included in the formulation adopted by the TCAD simulator [11], as derived from [12]. As discussed in [7], the effect of diffusion is assumed to be instantaneous, considering the phonon frequency is higher than that of the applied signal. The kinetic energy of the carriers under the electric field $E$ is limited by the effect of collisions. This is described by Euler's equation (7), where $\tau_p$ is the collision time.

Consider a sinusoidal voltage drop, $V_G(t) = \hat{V}_G e^{j\omega t}$, applied to the structure (with the positive sign at the gate). While the DC electric field is strictly confined within the depletion layer, the radiofrequency (RF) field may penetrate into the substrate, depending on the substrate doping and the value of the RF with respect to the dielectric relaxation frequency. At low frequencies, the region covered by the electric field is only the depletion region, extended by the equivalent thickness of the oxide $d_{eq}$ and by the Debye length, $\lambda_D = \sqrt{\frac{D_p}{\frac{\mu_p q N_A}{\epsilon_s}}}$. At high frequencies, the electric field penetrates the entire substrate, with a thickness $L_{sub}$. This result comes from the work of Blotekjaer and Quate [13], from which it can easily be inferred that the intensity of the RF electric field in the depletion layer is given by:

$$\hat{E} \cong -\frac{\hat{V}_G}{\frac{L_{sub}}{1+\frac{\mu_p q N_A}{j\omega\epsilon_s}} + \left(w + d_{eq} + \lambda_D\right)} \qquad (8)$$

In Equation (8), it is assumed that $L_{sub} \gg \left(w + d_{eq} + \lambda_D\right)$; it is also considered that the charge in the depletion layer can be neglected. With this last assumption, the amplitude of the first-order variations of electric field within the depletion layer does not depend on the position. As a direct consequence, this imposes a constant amplitude on the first-order hole velocity variation. The velocity variation can thus be considered constant in space and sinusoidal in time, superimposed onto the steady-state value: $v_p(t) = v_{p0}(x) + \hat{v}_p e^{-j\omega t}$. Euler's equation in the small-signal approximation can then be simplified as:

$$j\omega\hat{v}_p = -\hat{v}_p \frac{\partial v_{p0}}{\partial x} + \frac{q}{m_p}\hat{E} - \frac{\hat{v}_p}{\tau_p} \qquad (9)$$

In Equation (9), the collision time $\tau_p$ appears to be dependent on the electric field derivative. This quantity can be calculated by imposing the steady-state values of hole motility, assuming the equilibrium velocities into the depletion layer are $v_{p0}(x) = \mu_p E_0(x)$, where the steady-state electric field can be expressed as $E_0 = -\frac{q}{\epsilon}N_A x$. Thus:

$$\hat{v}_p = \frac{\frac{q}{m_p}\hat{E}}{j\omega + \frac{q}{\mu_p m_p}} = \mu_p'\hat{E} \qquad (10)$$

where $\mu_p'$ now appears as an equivalent, frequency-dependent mobility. This approach yields the same form as the Drude model. In the small-signal approximation, the first-order variation of the hole density has a sinusoidal time variation, which is added to the steady-state value: $p(x,t) = p_0(x) + \hat{p}(x)e^{j\omega t}$.

For the holes' continuity equation, including the known dependences of the steady-state distributions of the holes and the electric field, the following holds:

$$j\omega\hat{p} = -\frac{\partial\left[-\hat{p}\mu_p\frac{q}{\epsilon_s}N_Ax + p_{p0}e^{-\frac{\phi(x)}{V_T}}\mu_p'\hat{E}\right]}{\partial x} + D_p\frac{\partial^2\hat{p}}{\partial^2 x} \tag{11}$$

The steady-state potential in Equation (11) is described by Equation (2). In computing the derivative, it must be remembered that $\frac{\partial\hat{E}}{\partial x} = 0$ because the electric field variations are assumed to be constant in space within the depletion layer.

In particular, this last assumption means that Equation (11) presents one nonhomogeneous term with nonconstant parameters. A particular solution $\hat{p}(x) = \chi\frac{p_{p0}}{V_T}e^{-\frac{\phi(x)}{V_T}}\hat{E}x$ is required, where $\chi$ is unknown. By substituting, taking the derivatives, simplifying the term $e^{-\frac{\phi(x)}{V_T}}$, and collecting coefficients common to all terms, the following expression is obtained:

$$j\omega\chi = +\mu_p'\frac{q}{\epsilon_s}N_A - \frac{D_p}{V_T}\chi\frac{q}{\epsilon_s}N_A \tag{12}$$

leading to:

$$\chi = \frac{\mu_p'}{\frac{D_p}{V_T} + \frac{j\omega}{\frac{q}{\epsilon_s}N_A}} \ ; \ \hat{p}(x) = \frac{\mu_p'}{\frac{D_p}{V_T} + \frac{j\omega}{\frac{q}{\epsilon_s}N_A}}\frac{p_{p0}}{V_T}e^{-\frac{\phi(x)}{V_T}}\hat{E} \tag{13}$$

Equation (13) is the particular integral. A homogeneous equation can be obtained by canceling the nonhomogeneous term:

$$-\left(j\omega + \mu_p'\frac{q}{\epsilon_s}N_A\right)\hat{p} - \mu_p'\frac{q}{\epsilon_s}N_Ax\frac{\partial\hat{p}}{\partial x} + D_p\frac{\partial^2\hat{p}}{\partial^2 x} = 0 \tag{14}$$

Being a second-order equation, two solutions should be considered. Nevertheless, only the particular integral is considered for further calculations, for the following reasons:

(1) The coefficients of the first solution become null when the zero variation of hole density is imposed as a boundary condition at x = 0, where the depleted layer joins the substrate, which is populated by undepleted majority carriers;

(2) A second boundary condition considers the hole current variations to be zero at the silicon oxide interface. This implies that the coefficients of the second solution are negligible, as $\phi(w) > V_T$.

## 2.2. DC Equations

Hole density variations enter into the nonlinear term in Equation (6). The time average of the carrier fluxes gives rise to DC self-mixing terms, $J_{DC} = q\langle pv_p\rangle$, which introduce nonhomogeneous terms in the DC current balance equations. At zero frequency, the oxide capacitance means that the total current in the structure must be zero. Changes in carrier density caused by self-mixing are mainly due to the dynamics of the majority carriers, and therefore, recombination is neglected, and only the hole current equation is considered:

$$J_{DC} + qpv_p - qD_p\frac{dp}{dx} = 0 \tag{15}$$

Each of the quantities in Equation (15) can be considered, again, as the sum of a steady-state term plus a variation term: $p = p_0 + \widetilde{p}$, $v_p = v_{p0} + \widetilde{v}_p$, where $\widetilde{p}$ and $\widetilde{v}_p$ are the DC variation of hole density and hole velocity, respectively, induced in the depletion layer by the self-mixing effect. In determining the DC variations, unlike for the RF calculus, there is no external forcing generator; the first-order amplitude of variation of the electric field, $\widetilde{E}$, cannot be neglected. Assuming the equilibrium condition

among the steady-state terms and neglecting the higher-order terms, Equation (15) can be rewritten for small variations as:

$$
\begin{aligned}
J_{DC} &= -qp_0\widetilde{v}_p - q\widetilde{p}v_{p0}\ qD_p\frac{d\widetilde{p}}{dx} = \\
&= -qp_0\mu_p\widetilde{E} - q\widetilde{p}\mu_p E_0\ qD_p\frac{d\widetilde{p}}{dx}
\end{aligned}
\tag{16}
$$

The self-mixing process gives rise to variations in the electric field and hole density, part of which arise from variations in the electrostatic potential across the structure. Equation (16) does not seem to be analytically tractable. By adopting an "asymptotic" approach, it is assumed that only one of the three homogeneous terms has a dominant contribution, either in the hole or electron equation [7]. In this process, the two equations remain decoupled. The dominant term can be identified by evaluating the corresponding current contribution. For the hole current equation, the three solutions, indicated with subscripts A, B, and C, are given by:

$$
\widetilde{E}_A(x) = -\frac{J_{DC(x)}}{q\mu_p p_0}
\tag{17}
$$

$$
\widetilde{p}_B(x) = -\frac{J_{DC(x)}}{q\mu_p E_0}
\tag{18}
$$

$$
\frac{d\widetilde{p}_C(x)}{dx} = \frac{J_{DC(x)}}{qD_p}
\tag{19}
$$

Formally, these three terms generate current contributions, whose sum in Equation (16) must equal $J_{DC}$ under the same increment of the potential drop, $\widetilde{\phi}(x)$. As an asymptotic solution, the term which generates the smallest increment for the given self-mixing term is selected. Each contribution has the dimensions of an electric field; because the different terms in Equation (16) depict different physical effects, these contributions are defined with a particular symbol, $\widetilde{\phi}(x,y)$. The total potential $V_{mix}$ can then be obtained by integrating the selected contributions over the entire active area, $w$:

$$
V_{mix}(y) = \int_0^W \widetilde{\phi}(x,y)dx
\tag{20}
$$

The variable $y$ now indicates the position at which the potential is calculated, whereas $x$ indicates the position of the contribution to the potential. In the following, the subscript specifies the asymptotic solutions for A, B, and C. The terms $\widetilde{E}_A$ in Equation (17) correspond to local variations of the electric field when solution A is dominant. The corresponding local contributions to the potential are:

$$
\begin{aligned}
\widetilde{\phi}_A(x,y) &= -\widetilde{E}_A(x) = \frac{J_{DC(x)}}{q\mu_p p_0}\ for\ x \leq y \\
\widetilde{\phi}_A(x,y) &= 0\ for\ x > y
\end{aligned}
\tag{21}
$$

The terms of Equations (18) and (19) give rise to local variations in the hole density. The potential is thus obtained by integrating the electric field generated from this charge. Proper boundary conditions must be chosen in this case. To make the model result compatible with the TCAD simulation results presented later in this paper, the DC voltage is set to zero at the semiconductor body contact and at the gate contact. This corresponds to the condition imposed by the RF generator in the harmonic balance simulation and also to the steady-state condition in measurements, after the gate capacitance has been charged.

Any charge variation in the depletion region, which is necessary to sustain the diffusion in case B or C of Equations (18,19), results in the subtraction of charge from the contacts, partly from the bulk contact and partly from the gate contact. This effect forms a charge dipole. Related to the diffusion term, this feature is the effect referred to as self-mixing in a drift–diffusion equation.

Figure 2 illustrates a toy distribution composed of two deltas, $Q_A$ and $Q_B$. Using Gauss' equation, for any charge density variation, $Q(x)$ satisfies the following equilibrium condition:

$$E_1 x + E_2(w + d''_{ox} - x) = 0 \tag{22}$$

where $d''_{ox} = \frac{\varepsilon_S}{\varepsilon_{ox}} d_{ox}$ is the equivalent gate oxide thickness, and $E_1, E_2$ are the electric fields generated, respectively, at the body contact and gate contact by the charge $Q(x)$. The charge can be subdivided into $Q_1$ accumulated at the body and $Q_2$ accumulated at the gate, which can be expressed as:

$$Q_1 = Q(x)\frac{w + d''_{ox} - x}{w + d''_{ox}} \; ; \; Q_2 = Q(x)\frac{x}{w + d''_{ox}} \tag{23}$$

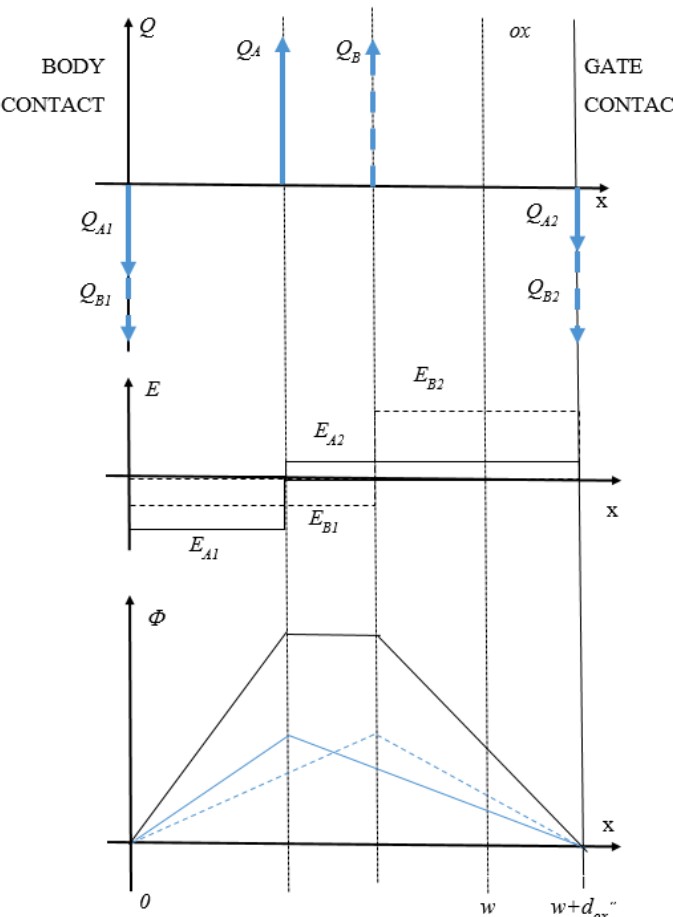

**Figure 2.** Schematic representation of self-mixing charge, electric field, and potential distributions.

For solution B, the local contribution to the potential is given by:

$$\widetilde{\phi}_B(x,y) = \frac{w+d''_{ox}-x}{w+d''_{ox}} y \frac{J_{DC(x)}}{\varepsilon_S \mu_p E_0} \; for \; x \leq y$$
$$\widetilde{\phi}_B(x,y) = \frac{w+d''_{ox}-x}{w+d''_{ox}} y \frac{J_{DC(x)}}{\varepsilon_S \mu_p E_0} \; for \; x \leq y \tag{24}$$

Integrating Equation (19) gives the variation in the carrier density:

$$\widetilde{p}_C(x) = \widetilde{p}_0 + \int_0^x \frac{J_{DC(z)}}{qD_p} dz \tag{25}$$

The boundary condition described at the border of the depletion layer imposes $\widetilde{p}_0 = 0$. For the third solution, the local contribution to the self-mixing potential is given by:

$$\widetilde{\phi}_C(x, y) = \frac{w+d''_{ox}-x}{w+d''_{ox}} y \frac{q}{\varepsilon_S} \widetilde{p}_C(x) \; for \; x \leq y$$
$$\widetilde{\phi}_C(x, y) = \frac{w+d''_{ox}-y}{w+d''_{ox}} x \frac{q}{\varepsilon_S} \widetilde{p}_C(x) \; for \; x > y$$

(26)

It is then simple to numerically integrate Equation (20).

This completes the description of the proposed model; note that this formulation is heavily reliant on an old self-mixing model that describes the acoustic–electric interaction [14].

## 3. Model Results

The developed model offers a comprehensive description of the vector dynamics within the field effect barrier and of photovoltage generation due to the self-mixing effect, allowing an assessment of the role of the different parameters. In this analysis, the gate voltage will be indicated as the voltage over the flat band condition, $V_{FB}$.

Figure 3 illustrates the distribution of the amplitude of hole current variations at three frequencies, namely 300 GHz, 1 THz, and 3 THz. The substrate p-type doping is $2 \times 10^{-17}$ cm$^{-3}$, the oxide thickness is 8 nm, and the gate voltage above the flat band is $V_G - V_{FB} = 0.17$ V. In this and all of the following simulations, an RF voltage of 1 mV is assumed. In particular, Figure 3 indicates that the RF oscillation of the carriers mainly occurs around the center of the depletion layer.

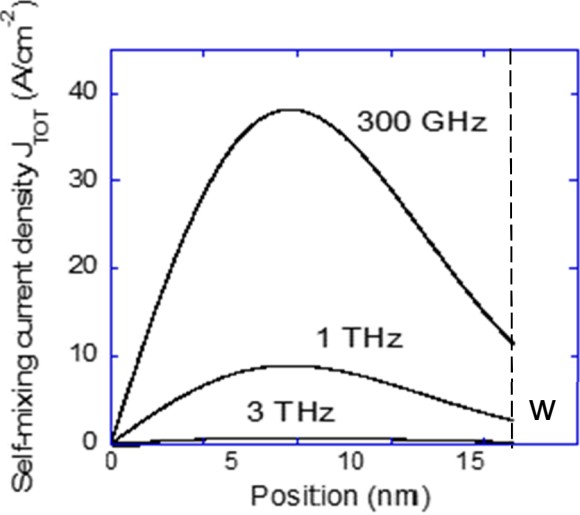

**Figure 3.** Distributions of the amplitude of hole current variations in the depleted layer at frequencies of 300 GHz, 1 THz, and 3 THz.

The variations in hole density and velocity, which appear within the nonlinear term of Equation (6), give rise to a nonhomogeneous term in the DC current, Equation (15). The solution of this equation is obtained using asymptotic approximations and gives three DC contributions to the potential, $\widetilde{\phi}_A$, $\widetilde{\phi}_B$, $\widetilde{\phi}_C$, which are shown in Figure 4. This figure illustrates the method used in the calculus, which consists of choosing the solution with the lowest potential contribution of the three. Different DC asymptotic approximations are indicated by dashed and dotted lines, and the continuous line indicates the resultant solution.

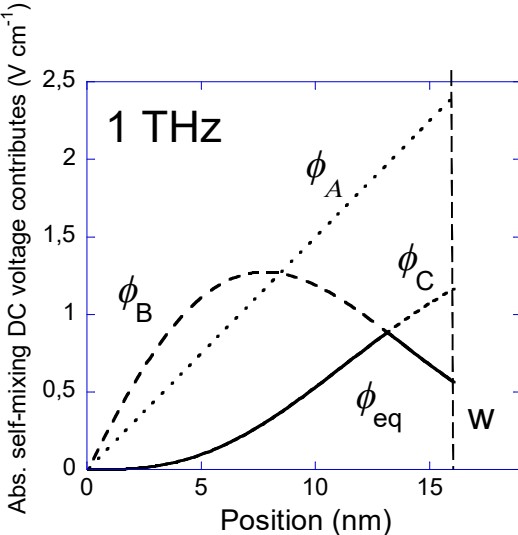

**Figure 4.** Distribution of the contributions of the different asymptotic terms (dashed and dotted curves) to the self-mixing voltage at 1 THz. Resulting contribution shown by the continuous curve.

The internal distribution of the photovoltage generated by the self-mixing effect is reported in Figure 5a, obtained from the model respectively at frequencies of 300 GHz, 1 THz, and 3 THz. As predicted, the maximum occurs within the semiconductor in the depletion layer, and the level decreases toward the oxide interface. In this case an extension of the substrate $L_{sub}$ =1.2 μm was assumed, in order to achieve a comparison with TCAD results, as reported in Figure 5b. This assumption does not change the shape of the potential distribution, only its amplitude, being related to the squared intensity of the electric field. In Figure 5b, the comparison is done at 1 THz, with $V_G$-$V_{FB}$ = 0.17 V. The curve obtained from the model is now translated with the origin at the oxide interface. The potential in the oxide has been added. One can observe how the shape of the distribution is maintained, including the moderate reduction of the peak potential toward the gate. Due to the abrupt depletion approximation, the model cannot follow the soft decrease of potential toward the substrate. At 1 THz, a relevant difference of the peak values can be observed, confirmed by the comparison of the Figures 7 and 9 introduced below. The author assumes this effect as due to the increase of mobility, considered by TCAD as an effect of ballistic behavior of carriers [15,16].

Figure 6 reports the dependence of the absolute value of the self-mixing photovoltage at the silicon dioxide interface ($x = w$) with respect to the gate voltage at three different substrate doping concentrations ($2 \times 10^{16}$, $2 \times 10^{17}$, and $2 \times 10^{18}$ cm$^{-3}$), all simulated at 1 THz. The photovoltage at the silicon oxide interface is proportional to the charge accumulated on the gate; thus, it can be assumed as an indication of a lock-in measurement.

Figure 7 reports the frequency behavior of the photovoltage at the same three doping concentrations of the substrate. Each simulation was performed using different values of the overdrive voltage, namely, 0.049, 0.074, and 0.17 V, respectively, for the concentrations $2 \times 10^{16}$, $2 \times 10^{17}$, and $2 \times 10^{18}$. As can be seen, at the lowest frequencies, the response tends to become constant, the lower doping giving the largest photovoltage. The response of the structure with lower doping also exhibits a lower cutoff frequency, so that at 1 THz, it is the highest doping that furnishes the largest photovoltage. This can be related to the variable confinement of the RF electric field within the depleted region described by Equation (8) and from the poles introduced by Equations (10) and (13).

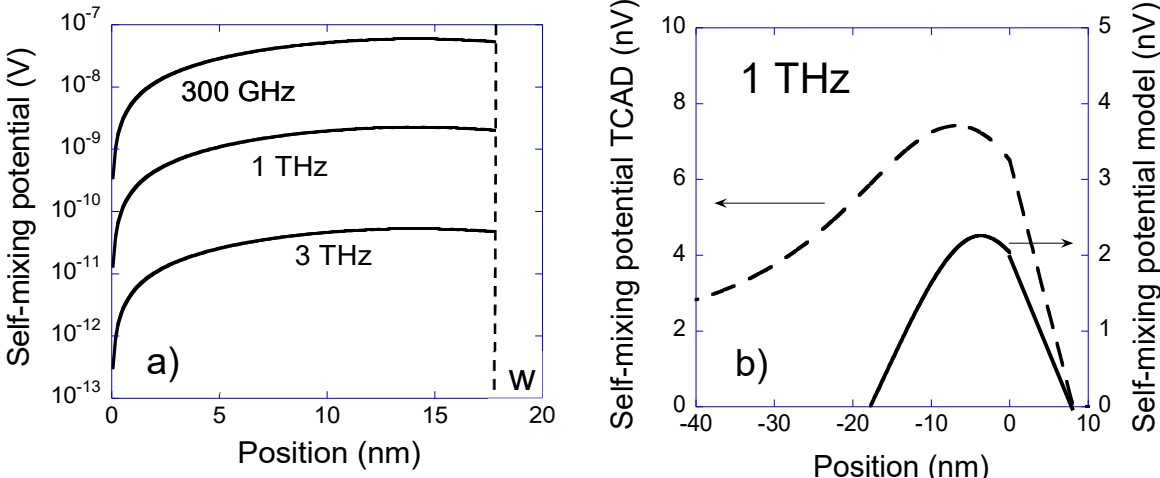

**Figure 5.** Distribution of the self-mixing potential in the depleted zone, (**a**) obtained from the model at frequencies of 300 GHz, 1 THz, and 3 THz, (**b**) comparison at 1 THz of results from Technology Computer-Aided Design (TCAD) simulation (dashed line) and from the model (continuous line).

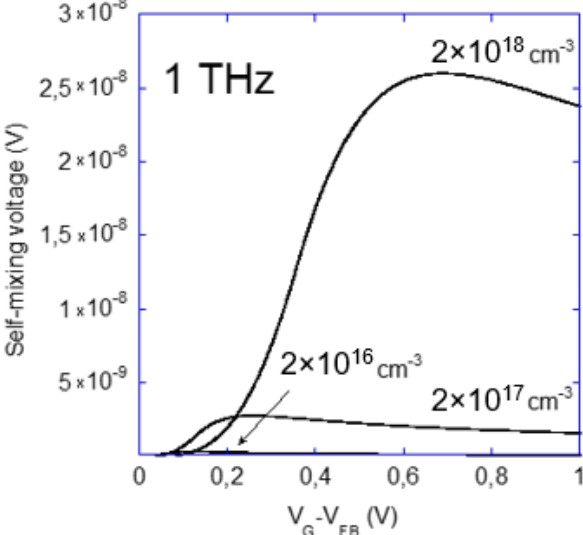

**Figure 6.** Absolute value of self-mixing voltage, calculated at 1 THz, for different values of the substrate doping, respectively: $2 \times 10^{16}$, $2 \times 10^{17}$, and $2 \times 10^{18}$ cm$^{-3}$.

As a model check, a detailed comparison with TCAD simulations is now presented. TCAD solves the semiconductor equations in the structure using the hydrodynamic equations. For these results, the harmonic balance tool was used, which furnishes the distributions of all the semiconductor quantities inside the structure at different harmonics. The zero-frequency term corresponds to the variations produced in the steady-state condition by the self-mixing. A simplified structure that approximates the one described by the model was analyzed. The structure is composed of a gate contact separated from the substrate by an 8 nm silicon dioxide layer. The substrate is homogeneously doped with boron at a density of $2 \times 10^{17}$ cm$^{-3}$. An Ohmic contact is placed at the bottom of the substrate, and the substrate has a thickness of $L_{sub} = 1.2$ μm. The gate in this structure is doped with $10^{20}$ cm$^{-3}$ phosphor. The flat band voltage in the structure is not zero; to maintain coherence with the model, the absolute value of the gate voltage is not specified, but the voltage above the flat band is set to 0.17 V. Between the gate and the body contacts, there is an RF voltage generator operating at a frequency of 300 GHz with 1 mV amplitude. The harmonic balance simulations were performed for five harmonics.

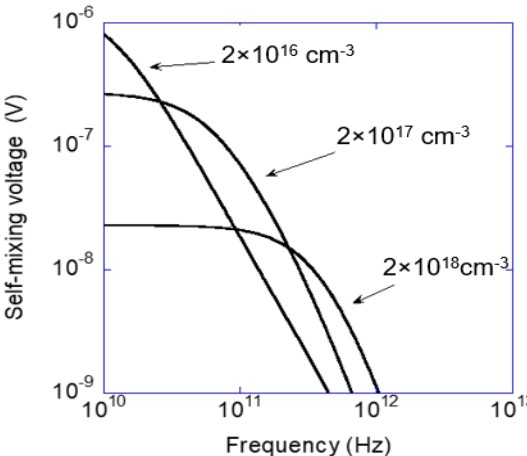

**Figure 7.** Frequency dependence of self-mixing potential in the metal oxide semiconductor (MOS) structure as obtained from the model, with three values of doping in the semiconductor body. Values of $V_G$-$V_{FB}$ are, respectively, 0.049 V, 0.074 V, and 0.17 V.

Figure 8a shows the 2D distribution of the DC potential arising from the self-mixing process just below the gate. Figure 8b reports the DC charge variation and shows the generation of a double layer of charge, constituting the dipole that gives rise to the photovoltage. Both figures use color maps and offer a pictorial representation of the effect produced by the RF field crossing the depletion layer. The relatively large dimension of the electrode means that the 2D picture provides a good approximation of the 1D model.

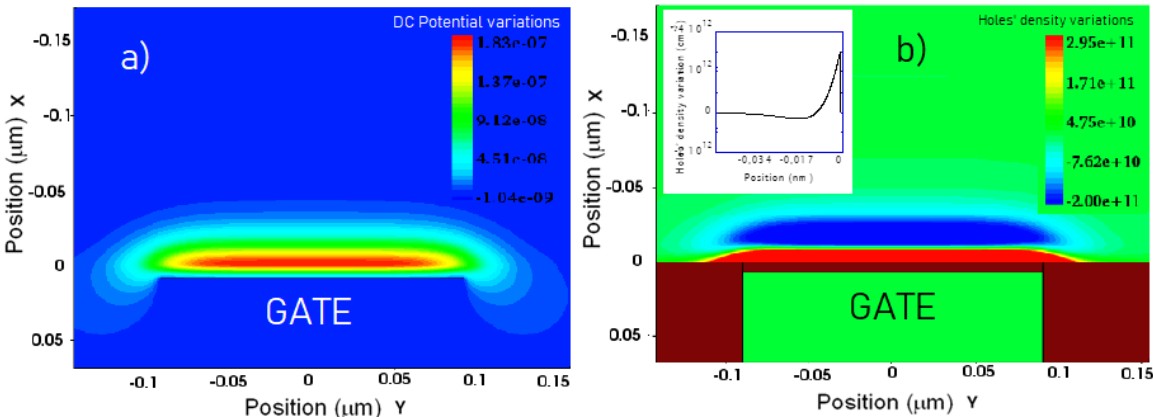

**Figure 8.** Results from the harmonic balance simulation of self-mixing. (**a**) Distribution of DC potential variation, the bar scale is Volts, extremes 1.8 $10^{-7}$/−1.0 $10^{-9}$ V; (**b**) distribution of DC charge variation, the bar scale is cm$^{-3}$, extremes $2.9 \times 10^{11}$/−$2.0 \times 10^{11}$ V; in this image, note the dipole, with positive (red) and negative (blue) charge variations. The inset reports a vertical cut at x = 0.

Figure 9 shows the harmonic balance simulation of the frequency dependence of the DC photovoltage at the silicon dioxide interface for three different doping values in the substrate: $2 \times 10^{16}$, $2 \times 10^{17}$, and $2 \times 10^{18}$ cm$^{-3}$. To obtain a comparison with Figure 7, the same overdrive voltages were assumed. The constant value at lower frequencies, with a cutoff frequency depending on the doping, strictly resembles the results obtained by the model. It should be noted that, in the two figures, the ends of the respective axes X and Y have been placed equal, in order to help compare the results and TCAD simulation results at 1 THz together. One can notice that that, at the highest frequencies, and depending on the doping, the TCAD simulated curves tend to saturate. The author assumes this effect as due to the increase of mobility, considered by TCAD as an effect of ballistic behavior of

carriers [15,16]. In these conditions the values obtained from the model, which consider a constant mobility, keep on decreasing.

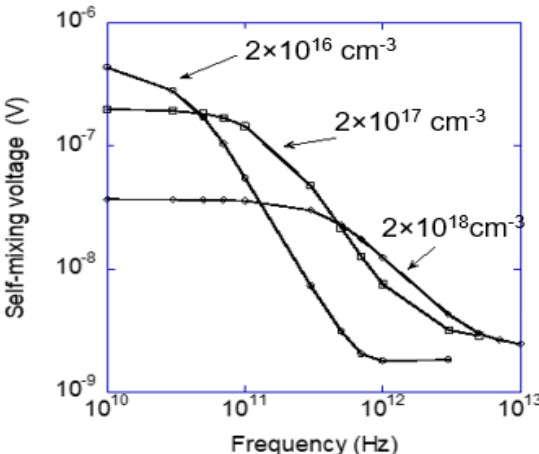

**Figure 9.** Frequency dependence of self-mixing potential in the MOS structure as obtained from TCAD simulations, with three values of doping in the semiconductor body. Values of $V_G - V_{FB}$ are, respectively, 0.049 V, 0.074 V, and 0.17 V.

We can add, as a final consideration of this section, the prediction of the responsivity which can be obtained from the model. First, we can extract only the value of the internal responsivity, since the antenna is not considered in this approach. Second, the model, among its innovative aspects, suggests that the photovoltage basically does not depend on the area of the capacitor, therefore on the power absorbed, as indeed the current does. Current responsivity thus has much more meaning than voltage responsivity. Considering the external circuit, the current should be detected during the charging transient, through a transimpedance amplifier, as a short circuit current.

Using the results of Figure 3, which show the photocurrent internally generated by the self-mixing effect, integrated along the depletion region, considering also the distribution of charge described by (23), we obtain $J_{SC} = 15\ \mu A/cm^2$. Using the AC conductance value of the structure, obtained at 1 THz from the TCAD simulation (2. $10^3\ \Omega^{-1}\ cm^{-2}$), the value of absorbed power, $P = 10^{-3}$ W, can be calculated. We obtain a theoretical intrinsic responsivity $R_I = 15$ mA/W, comparable with that found experimentally in [17].

## 4. Discussion of a Prospective 2D Model

The model presented in this paper is strictly 1D, as it focuses on an analytical description of the self-mixing process in the depletion region under the gate. Nevertheless, several aspects of this study suggest a possible extension to the 2D problem. Postponing an analytical approach for future study, this section qualitatively discusses such an extension.

The primary motivation for a 2D model is to seek an experimental confirmation of the modeling approach. Rather surprisingly, no THz-range photovoltage measurements on the MOS capacitance structure, i.e., between the gate and the body, have been reported in the literature. The trapping capacitance stimulated at high frequencies represents a different phenomenon [18]. This may, in part, be due to concerns that the antenna capacitance, connected to the gate, may dramatically reduce the detected voltage in transient lock-in measurements.

On the contrary, a large number of publications have reported measurements on THz rectification by MOS-FET [1,19–23] using the drain (sometimes the source) as the detection port, with the source (drain) electrode grounded. In such cases, the structure is 2D, and in principle a comparison with the 1D model should not be possible.

Nevertheless, the measurements reported in the literature all exhibit a bell-shape distribution of the DC potential with the gate voltage, very similar to the result obtained by the proposed 1D model at the gate. Additionally, the frequency behavior corresponds with the experimental results in the literature, even if the frequency response determined by the antenna must be taken into account in experiments [23]. One further comparison could consider the frequency response of a Schottky barrier [24], whose rectification effect, at very low RF voltage amplitude and high frequency, could be explained by the same self-mixing effect within the depletion barrier.

The hypothesis that arises from these observations is that the dielectric coupling between the self-mixing dipole and the gate, as described by the 1D model, may partially occur toward the drain and source doped zones. The barriers between the substrate and the doped zones avoid the flow of current. Nevertheless, in the presence of an electric dipole in the depleted layer, charge can be attracted to the edges of the respective depletion zones of the drain and source. During a chopped measurement, this effect would give rise to transient currents at the drain and source electrodes.

Figure 10 depicts the hypothetical situation, with the upper area, designed in light blue, schematically representing the depleted layer taken into account in the 1D model. The red region indicates an increase in the hole density, labeled Q following Figure 2. This region is beside the gate. It is dielectrically coupled with a region of decreased hole density, labeled Q1, at the edge of the depletion layer. The 1D model indicates that a negative charge Q2 is also induced at the gate. Capacitors represent the possible dielectric couplings, including in a 2D structure couplings occurring toward the source and drain areas. At this stage, this represents only a hypothesis that must be analytically evaluated, but it could justify the similarity between measurements at the drain/source and the model results. Following this conjecture, the bell-shaped curve of the measured photovoltage versus the gate voltage, similar to that obtained from the model, would be justified.

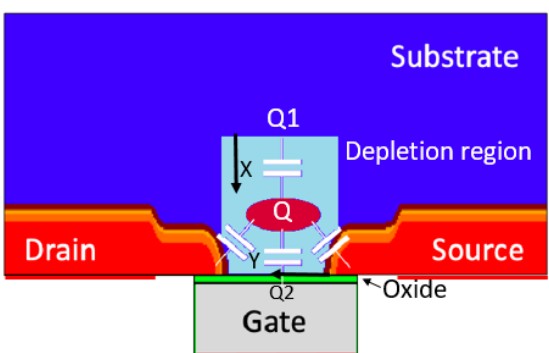

**Figure 10.** Schematic representation of the 2D MOS field-effect transistor (FET) structure showing the coupling paths of the charge in the self-mixing dipole. Orientations and origins of the two 1D models referred in the paper are also indicated.

For clarity, a classification related to the term "self-mixing" is now introduced. In general, "self-mixing" is defined the effect of the generation of a DC nonhomogeneous term due to the nonlinearity of the semiconductor current equation. The "self-mixing effect in a drift equation" is defined as the effect of the nonhomogeneous term evaluated in a current equation approximated as a single drift term, as in the case of the plasma wave model [2,5]. The "self-mixing effect in a drift–diffusion equation" is defined as the effect of the nonhomogeneous term evaluated in a current equation containing both drift and diffusion terms, as in the model presented in this paper.

A 2D representation of the structure would enable the differences between the model presented in this paper (developed along the X direction) and the plasma wave model (developed along the channel, in the orthogonal Y direction) to be identified. To this end, it would be useful to formalize the plasma-wave model using an equivalent circuit, as described in [2], and in particular Figure 1c in that

paper, where the most relevant assumption is clearly represented: the displacement current across the gate oxide becomes drift current in the channel.

Under this assumption, a large drift current flows along the channel, and gives rise, through the self-mixing effect in a drift equation, to the generation of a photovoltage. Note that the maximum photovoltage is generally measured below the threshold voltage at the gate. Thus, with limited carrier concentrations in the channel (values corresponding to dielectric relaxation frequencies well below 1 THz), the assumption of total conversion of the displacement current appears questionable.

In the 1D model presented in this paper, the RF screening effect of an eventual surface charge placed at the silicon oxide interface is ignored. The electric field proceeds from the gate oxide straight toward the depletion layer and possibly penetrates the substrate, as described in Equation (8). Under this assumption, the electric field gives rise to the generation of a photovoltage in the depleted layer. Following the above definition, this is a self-mixing effect in a drift–diffusion equation. The specification of two possible variants of the self-mixing effects highlights a further relevant difference between the two models. One direct consequence of the self-mixing effect in a drift–diffusion equation is the generation of a charge dipole under the conditions in which the diffusion terms become dominant, an effect that is confirmed by the simulations.

A future 2D analytical model should define the subdivision of the displacement current from the gate oxide and determine the relative influence of the two effects. The reason for the discussion presented here is to highlight which of these could be the role of the proposed 1D model.

## 5. Conclusions

This paper has presented a new model of the THz rectification process in an MOS structure. The model is based on the hydrodynamic semiconductor equations, solved for the small-signal approximation. The rectification effect is not limited by the damping effect of the parasitic capacitance but depends on the nonlinear effect of the carrier dynamics within the depletion layer, i.e., the self-mixing process. In particular, the model depicts the THz modulation mechanisms of the charge in the depleted regions of the structure. Numerical TCAD simulations have confirmed the model results. Thus, the model substantially improves the analytical models of self-mixing available in the literature and suggests a new interpretation of the detection process also in MOS-FET structures.

**Funding:** This research received no external funding.

**Acknowledgments:** The author wishes to thank Paola Loreti and Daniela Sforza for enlightening discussions on the differential equations.

**Conflicts of Interest:** The author declare no conflict of interest.

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
