# Peer review of "Self-Mixing Model of Terahertz Rectification in a Metal Oxide Semiconductor Capacitance"

_electronics, doi:10.3390/electronics9030479_

Round 1

Reviewer 1 Report

- Line 71 "At the same edge of depletion region, it is assumed that ohmic contact (body) occurs"
It is possible only for a very specific value of Vg.
Otherwise the equation (1) will be invalid.

- What is Lsub and where does equation (8) come from?

-Line 124 Square root is lost.

- Why do you drop the diffusion term of Euler's equation (9) in Eq.(10)?

- Incorrect sign in Eq.(15)

-Eq.16 is obscure

-Why the current density in Fig.3 in A/cm^3?

Reviewer 2 Report

I have reviewed the paper. These are my comments.

1) I think the authors should consult a professional editing service to render the English quality sufficient for publication.

2) The abstract and conclusion should be rewritten to reflect the important findings in the work.

3) In all figures, fond size of axis labels, axis values and legends must be sufficiently large for good reproduction; assume a 50% reduction of figure size in printed version.

4) Please add what is impact of this work in the future.

5) Please compare your work with similar work and what is the good point in your work. 

6) In Fig. 3, if we go below 300GHz, what gonna happen?

Reviewer 3 Report

In this paper the author presents a 1D model for self-mixing rectification in terahertz frequency region.

In my opinion the paper could have been written more systematically. Derivations are not easy to follow.

Some comments to the author:

1. It is not clear how can you get a DC current from the MOS capacitor (devices on Fig.1 and Fig.8). If there is no tunneling, there is no conductive path between gate and body electrode. In order to get DC current flow source and drain regions are needed. In my opinion to get rectification you need diode connected transistor or shunt capacitor between gate and drain to obtain self-mixing. Please, give comment. How can you get rectified voltage with device based on your model?

2. The structure from Fig.1 has a body contact at the edge of the depletion region. In text, it is claimed that Wtot can be calculated from (1). Does that mean that depletion region is calculated and then the contact is placed at the edge? Please clarify. It would be more convenient if Si/SiO2 interface is set at x=0, since now origin moves when you change any of the parameters of the model.

3. In section 2.1 give the names of the equations (Poisson, continuity, Euler). Where does the self-mixing come from? Is it the first term on the RHS in the continuity equation?

4. All the equations are solved in the x-direction. They should be solved in the direction of the DC current. If you place source and drain (2D) case, then they should be solved along the channel in y-direction as it is done in refs [2] and [4].

5. In the text : “At high frequencies, the electric field penetrates the entire substrate, with thickness Lsub.” Lsub is not given in Fig.1.

6. The first order variation of the electric field and the hole velocity gives the linearly increasing hole velocity? This could be written in a more simple way. In the 3rd paragraph on page 4, equation in text has vp(t) on the LHS and the RHS is function of f(x,t).

7. Equation (16) has problems with displaying mathematical operators. State clearly from which equations you obtain vp and p in (16).

8. Last sentence on page 6: “Each contribution has the dimensions of an electric field”.

It is not clear what you mean.

9. Solution of (16) by “asymptotic” approach includes some simplifications. How does it impact final solution? Comparison with numerical solution could be given. The calculation of Vmix(y) from PHI(x,y) is not obvious. Can you provide reference and the additional comment?

10. Please state from which equation follows the formation of charge dipole. Keep labeling consistent. Charge deltas in Fig.2 and text are labeled differently.

11. In the first sentence on page 8 reference the equations for B and C .

12. In Fig.3 check the units for current density. From which equation is current calculated? Give comparison with TCAD simulations.

13. Check the units for voltage in Fig.4. What does self-mixing voltage mean, is this the DC rectified component? The peak value is around 0.5V, and RF voltage is 1mV. Please explain the results.

14. On the inset of Fig.5 values at the y-axis are not visible. What is the difference between self-mixing DC voltage (Fig.4) and self-mixing potential (Fig. 5)?

15. Can you explain why you use term photo-voltage when you use RF voltage source for excitation both in model and TCAD simulations?

16. In the first sentence on page 11, the list of overdrive voltages is given. Which one is for which substrate concentrations.

17. The first sentence of the 2nd paragraph on page 11 – superscript 9 is given with no explanation.

18. Last paragraph on page 11. Show the 1D plot of the charge across the middle of the gate from Fig.8b. It would be easier to observe the charge dipole. Why the charge variations are plotted instead of charge? It would be best to see electrons and holes. Is this dipole result of the series combination of MOS and depletion capacitance?

19. Fig.7 shows the model and Fig. 9 TCAD simulations result, is that correct? It would be good to have both on the same plot.

20. Reference [1] and [17] are the same.

Reviewer 4 Report

The author reports on the theoretical description of a MOS capacitor for THz detection. The theory looks sound, however, requires some corrections in order to improve readability. There may also be a major bug (w_tot), but maybe after refinement, this becomes clarified. I recommend revisions, concerning the following list of comments:

-The paper clearly is about a MOS capacitor (and only a MOS capacitor!). The abstract and introduction, citing several papers on MOSFETs suggest that the paper also covers these, which is quite misleading. There are fundamental differences between the two. The most important one is that the carriers in a MOSFET originate from the source, while they are supplied by the body for a MOS-C. Also, the DC signal needs to be read out. The DS port, even of an ideal MOSFET, offers a non-infinite DC resistance which has strong impact on the noise equivalent power of the device. An ideal MOS-C, however, has infinite DC resistance (in reality possibly some GOhms). For THz frequencies, it will be almost purely imaginary, making impedance matching a difficult issue. Also the MOSFET concerns the change of potential along the gate (usually referred to the y-direction) while this paper describes rectification in the x-direction, perpendicular to the y-direction. Circuit effects have anyways been neglected in the paper… In order to make clear that the theory only considers a MOS-C, I recommend replacing the word “structure” by “capacitor” at least in the title.

-Eq. 1 and Fig. 1: I assume that w_tot is only the size of the depletion region and NOT the size of the body wafer. Fig. 1 is very misleading as the body contact is attached at x=0 and the size of w_tot starts from there. It looks like the whole body would be concerned. This is obviously wrong as the vast majority of the body wafer is uncharged (NA- and p0 are the same and cancel out any net charge). Please correct and name w_tot in the text.

-Eq. 10 is simply the Drude model which is a bit astonishing as the Drude model neglects the first term on the right hand side of Eq. 9. As v~E_0~x the derivative should survive and play a non-negligible role.

-The detection responsivity is missing! I fear that the detection efficiency is orders of magnitude lower than that of a MOSFET. The responsivity is one of the two parameters that is crucial to be able to determine whether the reported effect is a marginal contribution to THz detection (e.g. of a MOSFET) or whether it is an important part. From my own experience with FETs, I can tell that far below threshold (i.e. deep in the depletion) the response of the FET turns out to be very tiny, no matter whether the DS or the GS (=BS) terminal is read out. Unfortunately, the paper does not permit a direct comparison as nowhere in the paper any kind of power responsivity was calculated. The author assumes an AC bias of 1 mV but in order to transform this bias into power, the reader needs to know the incoupling resistance, e.g. that of an antenna. As the device impedance is basically purely imaginary, this bears challenges, e.g. considering impedance matching or similar (see e.g. https://iopscience.iop.org/article/10.1088/1361-6641/aae905/meta or https://ieeexplore.ieee.org/abstract/document/8371294 ). Therefore, the author should calculate the responsivity of the structure (photovoltaic mode might be best suited here) for a given toy example that could be justified.

-Ideally, the NEP should also be calculated.

-Fig. 2 may be a bit misleading; maybe rephrase to exemplary representation, as this is just a toy example of two charges (there are no two discrete charges in a real MOS-C)

-Fig. 5: A proper comparison of theory and simulation is required. The simulation unfortunately has arbitrary units (which generally should be omitted where possible). Therefore, it is not clear whether the order of magnitude of the effect is reproduced well. I recommend putting the simulation graph into the main figure with appropriate units.

-Fig. 7 and 9: please also put this in one figure for direct comparison of theory and simulation.

-Page 10, bottom: The author sates that “…proportional to the charge accumulated at the gate”. But the MOS-C is in depletion?! I do not understand this. I also don’t get what a “hypothetical lock-in measurement” shall be.

-Comparison to full 2D models: The MOSFET and HEMT rectification is very well understood with a vast majority of papers and pretty good agreement between theory and experiment. The so-called “bell-shaped curve” can be very well reproduced. Further, in most cases FETs are operated in the inversion region where this model does not hold (e.g Eq. 3 would not apply as it is the Boltzmann approximation; the full Fermi function must be used). There are also several investigations of FETs in the sub-threshold range (depleted channel) with quite some support by experimental data (e.g. https://www.nature.com/articles/srep20474.pdf or https://www.mdpi.com/1424-8220/18/11/3735). Further, there are some fundamental differences between a MOSFET and a MOS-C, see comment above. As the very same physical model (drift-diffusion equation and shallow water approximation) have been used in many papers on FETs already, I would think that the new theory provides little novelty there.

-Last but not least, circuit issues have been completely neglected. However, I agree that at this stage, such consideration are not necessary, they can be left for future work. What would be helpful, however, is the calculation of an ideal responsivity and compare that one to the ideal responsivity of FETs in the non-plasma-resonant mode in the long channel regime. A decent paper on this issue is e.g. https://aip.scitation.org/doi/full/10.1063/1.4826364

-Eq 10 : From Drude: q/m*µ=1/tau_p. Maybe both versions should be listed.

-Eq 20 ff: Useage of the greek letter phi for field is a bit misleading as phi is very frequently used for potential.

Formatting:

-In eq 7 a tabulator is missing (7)

-Two different symbols for the angular frequency omega have been used, c.f. Eq. 9 and Eq. 11,e g.

-Eq 16. The “+” sign in front of the third term is missing (twice)

-Fig.5: The inset is hardly readable. The theory units should not be a.u.

-Fig. 8: If printed black-white, the black numbers are hardly readable.

Round 2

Reviewer 1 Report

no comments

Reviewer 3 Report

1. OK.

You are detecting the charge transient which establishes the charge balance from Fig.2. Potential issues are that you are driving the input of the readout amplifier directly by the RF source and that you are detecting the small amplitude transient at the same time. Also, duration of the transient is important. It is not mandatory to address these issues, it can be left for the future work.

2. OK

3. OK

4. OK

5. OK

6. OK, but it seems to me that LHS should be vp(x,t).

7. OK

8. OK. But it would be good that at that point you gave short explanation (clarification) because the reader might be puzzled when looking at (18) and (19).

9. OK. But you should have given the reference point to [7] to make it clear. Comparison to numerical is fine.

10. OK

11. OK

12. OK. I think that vp and p can be visualized in TCAD.

13. OK.

14. OK

15. OK

16. OK

17. OK

18. OK

19. OK

20. OK.

Additional comment:

2.1. In line 74 – Beginning of the sentence has mistake.

2.2 Lines 233 and 234 – equation (24) has mistakes – one “epsilon_S” should be “q”

Reviewer 4 Report

I accept the corrections, the paper seems valuable, but there are some minor corrections still necessary:

Capacitance vs. MOSFET: in the abstract, 1st sentence, I would recommend adding: “…(MOS) capacitances within field effect transistors are…” as the named applications are MOSFET applications.

Eq. 10: I understood your derivation. But equation 10 is still the result of the Drude response, except that you write it down a bit differently. You should add a line, stating that your approach yields the same result as a Drude model.

New Fig.5: Either put all graphs in one figure or use the same scaling. Very difficult to compare. Also the quality of Fig. 5b seems low.

Fig. 7+9: my personal preference is the combined version in the cover letter. Or putting Fig. 7 and 9 next to each other with same scaling.

Section 2, 2nd sentence “Lsub” “sub” needs to be a subscript

There are still two different symbols used for omega, see Eq. 10 and the text below. Please correct.

Paragraph at the end of section 3: “We obtain a theoretical responsivity”-> “We obtain a theoretical intrinsic responsivity”. This is only the power coupled to the active element; you will never get this responsivity in a real experiment (circuit effects…)

Same paragraph: “Current responsivity thus has much more meaning than voltage responsivity”. I disagree. In fact, if you really had just a DC current reading (no modulation, e.g. by chopping), the steady state current MUST be zero (after a charge-up period): The currents calculated in the paper are currents towards the semiconductor-oxide interface (or away from it, depending on the substrate type, but always uni-directional!). As the (ideal) oxide has infinite resistance, it will charge up till there will be a counter-field that prevents further current to flow to the interface. The steady state current is therefore zero. But you can get a voltage build-up as the authors correctly answer to the question of reviewer 3. The reason why you might have gotten a current responsivity in an experiment is that you have employed some chopping. A lock in would just discriminate between a state of no current flow [chopper interrupts THz] and current flow towards the semiconductor-oxide interface [chopper open]. When THz beam is on, you will charge up the MOS capacitor, so a charging current will flow. During THz off, the current will flow back, neutralizing excess charge. So averaged over one cycle, there is indeed no net current flow, but as the lock-in measures the difference between these two states, you do indeed see a signal. Further, the Lock-in signal should heavily depend on the chopping frequency as the charge-up time is usually fast as compared to one chopping cycle. Extracting a current responsivity from this is questionable.

Same paragraph: I don’t understand how you got these values: In Fig. 3 you state that you used an RF voltage of 1 mV. Using P=1/2 sigma U^2 you get an (incredibly low) power of 65 fW/cm². Obviously not sensible... and it differs from your value. Also a value of 6.5E-8 /Ohm/cm² for the conductance sounds very unrealistic. I also wonder how you would couple THz power to such a structure.

Same paragraph, last line: “that found experimentally [A]”. What is [A]? Ref 8?
